# Reverse Genetics Systems for Emerging and Re-Emerging Swine Coronaviruses and Applications

**DOI:** 10.3390/v15102003

**Published:** 2023-09-26

**Authors:** Hui Jiang, Ting Wang, Lingbao Kong, Bin Li, Qi Peng

**Affiliations:** 1Nanchang Key Laboratory of Animal Virus and Genetic Engineering, Jiangxi Agricultural University, Nanchang 330045, China; pqjhpyx1103@163.com (H.J.); tingwang@jxau.edu.cn (T.W.); 2Institute of Pathogenic Microorganism, Jiangxi Agricultural University, Nanchang 330045, China; 3Department of Avian Infectious Diseases, Shanghai Veterinary Research Institute, Chinese Academy of Agricultural Science, Shanghai 200241, China; 4College of Bioscience and Engineering, Jiangxi Agricultural University, Nanchang 330045, China; 5Institute of Veterinary Medicine, Jiangsu Academy of Agricultural Sciences, Key Laboratory of Veterinary Biological Engineering and Technology, Ministry of Agriculture, Nanjing 210014, China

**Keywords:** emerging and re-emerging swine coronaviruses, reverse genetics system, pathogenesis, vaccine development, antiviral drugs, cell and tissue tropism

## Abstract

Emerging and re-emerging swine coronaviruses (CoVs), including porcine epidemic diarrhea virus (PEDV), porcine deltacoronavirus (PDCoV), and swine acute diarrhea syndrome-CoV (SADS-CoV), cause severe diarrhea in neonatal piglets, and CoV infection is associated with significant economic losses for the swine industry worldwide. Reverse genetics systems realize the manipulation of RNA virus genome and facilitate the development of new vaccines. Thus far, five reverse genetics approaches have been successfully applied to engineer the swine CoV genome: targeted RNA recombination, in vitro ligation, bacterial artificial chromosome-based ligation, vaccinia virus -based recombination, and yeast-based method. This review summarizes the advantages and limitations of these approaches; it also discusses the latest research progress in terms of their use for virus-related pathogenesis elucidation, vaccine candidate development, antiviral drug screening, and virus replication mechanism determination.

## 1. Introduction

Coronaviruses (CoVs) are the largest group of positive single-strand RNA viruses in the *Coronaviridae* family, and they are represented by four genera: *Alphacoronavirus*, *Betacoronavirus*, *Gammacoronavirus*, and *Deltacoronavirus.* All CoVs have similar genomic organization; they have a genome that is approximately 26–32 kilobases (kb) in length and contains at least six open reading frames (ORFs): ORF1a, ORF1b, spike (S), envelope (E), membrane (M), and nucleocapsid (N) [1]. Two overlapping ORFs, ORF1a and ORF1b, encode the polymerase proteins, which are further cleaved into nonstructural proteins. The S protein is responsible for the binding of the virus to specific host receptors, enabling its entry into host cells; it is also the main structural protein for eliciting viral neutralization antibodies [2,3,4]. E and M proteins are the most abundant and are the most conserved membrane proteins in CoVs; as such, they are often used as the target proteins to establish diagnostic tools [5]. N protein forms a complex with genomic RNA and interacts with M protein during virion assembly [6].

Thus far, six CoVs have been reported to infect pigs: transmissible gastroenteritis virus (TGEV), porcine respiratory CoV (PRCV), porcine epidemic diarrhea (PED) virus (PEDV), swine acute diarrhea syndrome-CoV (SADS-CoV), porcine hemagglutinating encephalomyelitis (PHEV), and porcine deltacoronavirus (PDCoV). TGEV, PRCV, PEDV, and SADS-CoV belong to the genus *Alphacoronavirus*, PHEV belongs to *Betacoronavirus*, and PDCoV belongs to *Deltacoroanvirus*. PEDV, SADS-CoV, and PDCoV are considered emerging or re-emerging swine CoV, whereas TGEV, PRCV, and PHEV have been known to infect pigs for decades. PEDV was first identified in Europe in the early 1970s [7]; however, it re-emerged in the late 2010s, when it caused significant economic losses to the swine industry worldwide [8,9]. PED transmission was well controlled in China during the 1990s and early 2000s because of the widespread use of CV777-based vaccines (inactivated or live-attenuated vaccine) [10]. However, the emergence of prevalent mutant PEDV strains (such as highly virulent PEDV variants) has limited the protection efficiency of these vaccines, resulting in large PED outbreaks in China [11,12]. In many pig-raising countries, PED outbreaks continue to cause substantial economic losses to the swine industry [13,14,15]. PDCoV was first identified in 2012 from porcine fecal samples in a Hong Kong surveillance study [16]. Thereafter, a high prevalence of PDCoV infection among pigs was reported in the United States and China, and it was associated with certain economic losses to the swine industry [17,18]. SADS-CoV, a bat-origin alphacoronavirus, causes severe diarrhea, vomiting, dehydration, and even death in pigs [19]. SADS-CoV was first detected in four pig farms in Guangdong, China, in 2017, resulting in the death of 24,693 piglets [20]. Later, SADS-CoV was identified in Fujian and Guangxi in 2018 and 2021, respectively [21,22]. Because SADS-CoV infection is associated with a high mortality rate in piglets, there is an urgent need for molecular surveillance of SADS-CoV in swine farms.

Viral reverse genetics involves modifications intentionally introduced to the genome at the viral genome level, including mutations, deletions, insertions, and substitutions. Reverse genetics techniques are beneficial for studying viral gene functions, screening antiviral drugs, and developing vaccines; however, because CoVs have a large genome and a replicase gene sequence that is unstable during cloning in bacteria, establishing a CoV reverse genetics system is difficult. Thus far, several methods, including targeted RNA recombination, in vitro ligation, bacterial artificial chromosome (BAC)-based ligation, vaccinia virus-based recombination, and yeast-based technology, have been applied to construct reverse genetics systems for swine enteric CoVs. In this review, we summarize the established reverse genetics systems for the emerging and re-emerging swine CoVs and describe their applications to understand the effects of CoV proteins on viral virulence and innate immunity, cell and tissue tropism, transcription regulatory sequence activity determination, and antiviral drug screening.

## 2. Reverse Genetics Systems for Swine Enteric CoVs

### 2.1. Targeted RNA Recombination

Targeted RNA recombination was the first technique used for constructing a reverse genetics system for recombinant PEDV in 2013. It is based on the high homologous RNA recombination rate in CoVs and the cell tropism determinant of S protein [23]. This technique has also been applied to other CoVs, including feline infectious peritonitis virus (FIPV) [24], infectious bronchitis virus (IBV) [25], and murine hepatitis virus (MHV) [26]. The system operation of PEDV is divided into two stages (Figure 1): First, MHV S mRNA containing the upstream and downstream homologous arms of the PEDV S gene is transcribed from the T7 promotor and electroporated into PEDV-infected Vero cells. Then, the cells are overlaid onto a murine cell (L cell) monolayer. The recombinant mPEDV is generated during subsequent incubation after two rounds of plaque selection on L cells. Second, to recover the recombinant virus, the expected mutant transcripts with homology arms are electroporated into mPEDV-infected murine L cells. Then, the infected L cells are overlaid onto a monolayer of Vero cells to rescue the recombinant virus. The recombinant virus is obtained after two rounds of plaque purification in Vero cells. Although targeted RNA recombination is the first established reverse genetics system for swine CoVs, manipulating two-thirds of the 5′ end of the CoV genome can be difficult. No tedious cloning procedure of the whole CoV genome is the main advantage of this approach. However, several rounds of plaque purification are needed to obtain the recombinant CoV, which makes this method time- and labor-intensive.

### 2.2. In Vitro Ligation Method

Due to the toxicity and instability of the CoV replicase gene in the bacterial cloning process, some researchers have ligated the full-length CoV genomic cDNA directly in vitro and then obtained the infectious mRNA through transcription. Here, a recombinant virus can be obtained through electroporation of the infectious mRNA into susceptible cells. This method of ligating full-length CoV cDNA relies on type II restriction endonucleases, such as *Van*91I, *Bg1*I, *Sap*I, *Bst*XI, *Bsm*BI, *Aar*I, and *Bsa*I. These restriction endonucleases cleave DNA fragments to leave sticky ends with random bases, ensuring that individual fragments are ligated only directionally. After the ligated cDNA is purified, the infectious mRNA can be obtained through in vitro transcription. Next, the recombinant virus can be obtained by electroporating this mRNA into susceptible cells. Reverse genetics systems based on in vitro ligation have been successfully established for many CoVs, including MHV, PEDV, IBV, severe acute respiratory syndrome CoV (SARS-CoV), human CoV NL63, Middle East respiratory syndrome-related CoV (MERS-CoV), and SARS-CoV-2 [27,28,29,30,31,32].

In 2016, Beall et al. established a reverse genetics system for PEDV strain PC22A [33]. For constructing the infectious clone of PC22A, the authors divided the PEDV genome into six segments for subcloning (Figure 2). The individual segments were enzymatically cleaved, purified, and ligated. The assembled full-length cDNA, containing a T7 RNA polymerase promoter at the 5′ end and a poly(A) tail at the 3′ end, was transcribed in vitro to generate capped full-length transcripts co-electroporated with capped N gene transcripts to efficiently recover an infectious virus (Figure 2). At 12 h post-electroporation, the medium was replaced with DMEM containing 5 μg/mL trypsin to facilitate virus recovery and spread. The in vitro ligation is the commonly used method in the construction of CoV’s reverse genetics system because it does not need to clone large DNA fragments in one vector and has the high recovery efficacy of a recombinant virus. However, appropriate type II restriction endonuclease sites are needed to check in CoV’s genome to ensure directional assembly of CoV genome.

### 2.3. BAC-Based Ligation

The CoV genome is large (26–32 kb) and has a replicase gene with cDNA sequences, which are unstable in bacterial cloning systems. The BAC vector is based on the bacterial F’ factor, which regulates the DNA synthesis so that its copy number is kept at a low level; this ensures the stability of its cloned genes and increases its tolerance for >300-kb genes [34]. As such, many researchers have applied BAC plasmids to effectively construct reverse genetics systems for CoVs, including SARS-CoV, MERS-CoV, FIPV, TGEV, MHV, and PEDV [35,36,37,38]. For instance, we constructed an infectious cDNA clone for the PEDV strain AJ1102 using the pBeloBAC11 as a cloning vector in 2020 [39]. First, we constructed an intermediate plasmid pBAC-M-PEDV, which cloned the cytomegalovirus (CMV) promoter sequence, PEDV genome segments [nucleotide (nt) 1–1,092, *Pac*I cleavage site, followed by nt 22,199 to 3′ untranslated region (UTR)], a poly(A) tail, hepatitis delta virus ribozyme (HDVr), and a bovine growth hormone (BGH) polyadenylation sequence into the pBeloBAC11 vector. The remaining AJ1102 genome was divided into five fragments and amplified through a reverse transcription (RT) polymerase chain reaction (PCR). The five DNA fragments were assembled into a pBAC-M-PEDV vector through homologous recombinations to obtain an infectious clone of AJ1102. The recombinant virus could be recovered through transfection of 6 μg of the recombinant BAC plasmid into Vero cells and by replacing the medium with DMEM containing 5 μg/mL trypsin 6 h after transfection (Figure 3).

The BAC system has also been applied to other swine CoVs. For example, Zhou et al. constructed an infectious clone for PDCoV CHN-HN-1601 using a BAC plasmid as the cloning vector [40]. The authors used pBeloBAC11 to construct an intermediate plasmid, pBeloBAC11-M, cloning a synthesized gene including the CMV promoter, a partial N-terminal region of the viral genome (nt 1–2,869), a partial region of ORF1b (nt 11,591–11,687) containing a *Bst*BI restriction site, a partial C-terminal region of PDCoV genome (nt 24,030–25,419), a 25 nt poly(A) tail, HDVr, and a BGH sequence. Then, five DNA fragments encompassing the genome of PDCoV CHN-HN-1601 were amplified through RT-PCR, and the first two DNA fragments were ligated into pBeloBAC11-M by seamless cloning using an NEBuilder HiFi DNA Assembly Cloning kit (New England BioLabs, MA, USA). Similarly, the other three DNA fragments were also ligated into pBeloBAC11-M through seamless cloning. Finally, the infectious clone plasmid can be obtained through restriction digestion and ligation with *Bst*BI and *Mlu*I. Virus recovery can be performed through transfection of recombinant BAC plasmid into LLC-PK1 cells, followed by replacement of the medium with 10 μg/mL trypsin and 37.5 μg/mL pancreatin. The limitation of this approach is that tedious cloning procedures are needed to obtain a recombinant BAC plasmid encoding the full-length CoV genome. Due to the low copy number of BAC in a bacterium, extracting a certain amount of BAC plasmid from bacteria is time-consuming. However, several advantages, including high stability of exogenous genes, high efficacy of cDNA transfection into susceptible cells, and easy manipulation for genes, make it is an attractive method to establish CoV reverse genetics system [41].

### 2.4. Yeast-Based Method

An advantage of using the yeast system for constructing a CoV reverse genetics system is that >10 cDNA fragments can be ligated simultaneously such that the total fragment length exceeds 110 kb. In 2022, Zhou et al. reported the construction of a PEDV reverse genetics system using the yeast system [42]. The sequences of CMV promoter, HDVr, and BGH termination signal sequence were cloned into pYES1L vector to obtain the vector pYES1L-CMV-HDVrbz-BGH. DNA fragments encompassing the whole genome of PEDV HM strain were amplified through RT-PCR with fidelity polymerase. Next, these DNA fragments were ligated into pYES1L-CMV-HDVrbz-BGH through transformation-associated recombination (TAR) in yeast to obtain pYES1L-PEDV, the HM strain infectious cDNA clone (Figure 4). The infectious virus could be recovered through the transfection of pYES1L-PEDV into Vero cells. Rapid assembly of the whole CoV genome in one time is the main advantage for yeast-based method. However, this method needs extract yeast plasmid from yeast culture; yeast plasmid extraction is more complicated, time-consuming, and expensive than bacterial plasmid extraction [42].

### 2.5. Vaccinia Virus-Based Recombination

As a cloning vector, the vaccinia virus enables stable propagation of full-length CoV genomic cDNA in cell culture, as well as genetic manipulation through vaccinia virus-mediated homologous recombination (Figure 5). Kristen-Burmann et al. constructed an infectious clone of PEDV by using the vaccinia virus as a cloning vector and evaluated the pathogenic role of the ORF3 and S gene in the PEDV US strain MN [43]. The genome of PEDV-MN was divided into eight cDNA fragments (F1–F8), which were then used to generate four plasmids: pA, containing fragment 1 upstream and fragment 8 downstream of the guanosine phosphoribosyl transferase (GPT) gene; pB, containing fragment 2 and 7; pC, containing fragment 3 upstream and fragment 6 downstream of the GPT gene; pD, containing fragment 4 and 5. The introduction of the full-length PEDV-MN genomic cDNA into the vNotI/tk vaccinia virus genome was performed through four rounds of vaccinia virus-mediated homologous recombination by using GPT as a positive or negative selection marker. In brief, vaccinia virus-mediated homologous recombination was carried out as follows: CV-1 cells (5 × 10^5^) were infected with the respective recombinant vaccinia virus at 1 multiplicity of infection; at 1 h after infection, 5 μg of plasmid DNA was transfected using Lipofectamine 2000 (Invitrogen). After 2 days, plaque purification was performed to isolate the recombinant vaccinia virus after three rounds of plaque purification with GPT-positive or -negative selection as a selection marker.

However, even after the full-length cDNA of PEDV-MN was cloned into vNotI/tk vaccinia virus genome, an attempt to recover the virus failed. The recombinant PEDV was successfully recovered only after the authors modified the genome by substituting two nucleotides within the 5′UTR and two additional nucleotides within the S gene based on the sequence of cell-adapted PEDV CV777 [43]. The vaccinia virus genome has many non-essential regions, which enable the insertion of large fragments of foreign genes without affecting viral replication. Vaccinia virus can be easily cultured to high titers and can effectively infect primary, mammalian, and non-mammalian cell lines. However, several rounds of vaccinia virus-mediated homologous recombinations and plaque purifications make this method more complicated and time-consuming.

### 2.6. Alternative Methods

#### 2.6.1. Transformation-Associated Recombination Cloning

A 2020 study from Switzerland and Germany used a transformation-associated recombination (TAR) cloning platform to construct a CoV reverse genetics system. The authors constructed a recombinant novel CoV SARS-CoV-2 virus using the GFP protein to replace ORF7a, an accessory protein [44]. First, DNA fragments with 45–500 base pair (bp) overlaps can be obtained by overlapping PCR or RT-PCR of viral RNA extracted from viral strains. The vector DNA fragment used for TAR cloning was amplified through PCR from vector pCC1BAC-His3 with 45-bp overlaps to fragments encompassing the 5′ or 3′ ends of different viral genomes. Next, for viral genome assembly, all DNA fragments were transformed into *Saccharomyces cerevisiae* VL6-48N. This was followed by the extraction of yeast artificial chromosome (YAC) from the yeast culture medium and cleavage of YAC-containing viral cDNA at the unique restriction site downstream of the 3′-end poly(A) tail. After transcription in vitro, the infectious mRNA was electroporated along with the N gene transcript into susceptible cells to recover the recombinant virus (Figure 6). Although this method has not yet been applied to swine CoVs, it enables manipulation of the CoV genome without multiple subcloning steps and allows for rapid recovery of the recombinant virus. However, the efficacy of the whole viral genome assembly in yeast requires further analysis.

#### 2.6.2. Circular Polymerase Extension Reaction-Based Reverse Genetics System

The recently reported circular polymerase extension reaction (CPER)-based method may be an attractive method for recombinant-positive RNA virus recovery [45]. CPER is bacteria- or yeast-free and does not require complicated cloning procedures for establishing an infection system. However, it requires a viral genome obtained through the amplification of the overlapping DNA fragments encompassing the viral genome using a fidelity PCR polymerase. A linker DNA fragment, containing homology arms, CoV 3′ UTR, a poly(A) tail, HDVr, a BGH polyadenylation sequence, a spacer sequence, a CMV promoter sequence, and CoV 5′ UTR, was also amplified through fidelity PCR. A single CPER using a high-fidelity DNA polymerase can be performed in one PCR tube with the amplified DNA fragments as templates; it yields circular DNA containing the full-length viral cDNA (Figure 7). Then, the CPER mixture was transfected into susceptible cells to recover the recombinant virus [46]. Thus far, CPER has not yet been used for swine CoVs. Nevertheless, Torii et al. constructed a reverse genetics system for SARS-CoV-2 using a CPER-based method and successfully inserted two reporter genes, sfGFP and HiBiT luciferase genes, into the genomic sites of ORF7a and ORF6 genes, respectively [47]. CPER is an easy and rapid method to manipulate such a large viral genome, making it an attractive approach for rapid recovery of positive-strand RNA viruses. However, the efficacy of this approach for obtaining recombinant viruses with expected mutations warrants evaluation.

## 3. Application of Reverse Genetics Systems to Swine Enteric CoVs

### 3.1. Virulence Determinant Identification

Identification of the virulence factors in CoVs is essential for the rational design of live-attenuated vaccine candidates. To date, an extensive effort has been made toward understanding the virulence factors of the emerging and re-emerging swine CoVs [48,49,50,51,52,53,54,55]. Many studies have suggested that the virulence of PEDV is associated with its S protein. In 2017, Hou et al. constructed a recombinant PEDV with a 197-amino-acid deletion in the N-terminal domain of the S1 subunit; their animal experiment results demonstrated that the deleted 197-amino-acid is a virulent factor in PEDV [55]. The reverse genetics systems have also been applied to evaluate the role of the S1 subunit between different genotypes. Chimeric PEDVs with the S1 subunits from different genotypes were generated, and the results demonstrated that the S1 subunit contributes to differences in virulence between PEDV G1 and G2 strains [56,57]. Hou et al. also constructed a recombinant PEDV with a deleted YxxF motif at the C-terminus of the S protein; the recombinant PEDV was noted to reduce virulence in pigs [48]. These studies suggested that the S2 subunit is also an important virulence factor for PEDV. In 2018, Wang et al. applied the reverse genetics systems of two PEDV strains to confirm that the S gene is not the only virulent factor in PEDV [49]. Li recently deleted seven amino acids (aa 23–29) in PEDV E protein, and this deletion can attenuate PEDV but retain its immunogenicity and promote IFN expression [58].

In addition to structure proteins, nonstructural proteins, including nsp1, nsp15, and nsp16, are also linked to the virulence of PEDV. Niu et al. mutated two amino acids (N93A and N95A) in PEDV nsp1 and noted that the recombinant virus increased viral sensitivity to the host immune response and demonstrated an attenuation phenotype in vivo [52]. We also found that complement component 3 (C3) significantly inhibited PEDV replication in vitro; however, PEDV can antagonize the immune suppression of C3 via inhibition of the nsp1 phosphorylation of CEBP/β [59]. Mutation of the residue V50 in nsp1 can attenuate the immune evasion effects of variant PEDV. In 2020, Deng et al. inactivated three interferon antagonists (nsp1, nsp15, and nsp16), thus attenuating PEDV pathogenesis [53]. At least 11 viral proteins encoded by PEDV can inhibit the IFN responses, including nsp1, nsp3, nsp5, nsp7, nsp14, nsp15, nsp16, ORF3, E, M, and N [60,61,62,63,64,65,66,67]. Although the mechanisms underlying the inhibition of IFN responses by these viral proteins vary, inactivating any of the IFN antagonist sites in these proteins may attenuate PEDV pathogenesis. The nsp16 of PEDV encodes the 2′-O methyltransferase, which is responsible for the methylation of the 2′-O in the first ribose of viral RNA. Hou et al. suggested that the inactivation of 2′-O methyltransferase activity in the PEDV strain PC22A can reduce its virulence [51] and provide sufficient protection until three weeks after the viral challenge. Deng et al. inactivated the nsp15, which encodes the endoribonuclease, to obtain an attenuated PEDV; the recombinant virus could promote the expression of both type Ⅰ and Ⅲ IFN [54]. Studies have also focus on identifying the virulence factor in PDCoV. In 2019, Zhang et al. found that piglets inoculated with NS6-deleted PDCoV did not show any clinical signs of infection, indicating that the accessory protein NS6 is an important virulence factor for PDCoV. However, NS6 deletion in PDCoV would reduce the viral titers in vivo and in vitro [50].

### 3.2. Rational Design Vaccine Candidates

The aforementioned virulence factors for swine CoVs can be the target for manipulation by reverse genetics systems for rapid virus attenuation. The attenuated virus particles can then serve as attenuated vaccine candidates. By using the reverse genetics system of PEDV AH2012/12, we recently proved that the C-terminus of the S2 subunit contributes to PEDV virulence. We further confirmed that compared with the killed vaccine, the recombinant PEDV with seven-amino-acid deletion at the C-terminus of S protein elicits increased immunoglobulin (Ig) G and IgA, neutralization antibody production, and better protection effects against virulent virus challenge.

The efficient replication of many swine CoVs, such as PEDV and PDCoV, requires the presence of trypsin. By using the reverse genetics system, some studies have identified the S protein of PEDV as the determinant of this trypsin dependency [68,69,70]. Substitution of the S2_720–892 aa_ with a trypsin-independent strain or the furin cleavage site in S protein induces trypsin independence during PEDV replication. Notably, replacement of S2_894–1386 aa_ in G2 PEDV with the trypsin-independent G1 strain can confer protective effects against both G1 and G2 PEDV challenges and enable propagation to high viral titers in vitro. Changing the biological characteristics of the swine CoVs, such as adding trypsin independence and increasing viral titer in vitro by using reverse genetics systems, can improve the swine CoV vaccine manufacturing processes.

Moreover, the reverse genetics system has the potential to be an expression vector that delivers antigens of interest. For example, Pascual-Iglesias et al. generated a recombinant chimeric TGEV that expresses PEDV S protein, which could protect against challenges with a virulent PEDV strain [71]. Because their study lacked animal experiments with a TGEV challenge, it is still unknown whether the chimeric TGEV can protect against virulent TGEV challenge. Li et al. engineered a bivalent vaccine using PEDV YN150 as the backbone to express VP7 of porcine rotavirus, and vaccination of the piglets with the recombinant virus protected them against both PEDV and porcine rotavirus infection [72].

### 3.3. Delineation of Cell and Tissue Tropism

Binding to cognate receptors is a prerequisite for the initiation of virus infection. For instance, the binding of S protein from SARS-CoV-2 to different species’ angiotensin-converting enzyme 2 (ACE2) is the molecular basis of the broad host range for SARS-CoV-2 [73,74]. We previously noted that the receptor binding domain (RBD) of PDCoV binds to the conserved residues of aminopeptidase N (APN) of different species, which exhibits a high risk for cross-species transmission [75]. PEDV is an important swine CoV that continues to cause economic losses to the swine industry. Isolation of wild PEDV in vitro remains challenging; this greatly impedes the PEDV vaccine development process. A study by Li et al. found that the S gene is the determinant for the adaptation of PEDV in both LLC-PK1 and Vero cells [76]. Using a reverse genetics system, the authors discovered that the S1 subunit and half of the S2 are critical for the cellular adaptability of PEDV. Similarly, Chen et al. found that three amino acid mutations (A605E, E633Q, and R891G) in S protein enable the attenuated PEDV strain DR13 to efficiently replicate in Vero cells [77].

CoV infection is initiated by the interaction between S protein and specific cellular receptors; this biological process largely determines a CoV′s host spectrum and tissue tropism. Phylogenetic analysis reveals that PDCoV is closely related to sparrow CoV HKU17, which supports the hypothesis that PDCoV evolved from avian deltacoronavirus [16,78]. Niu et al. constructed chimeric PDCoVs that harbor the S protein of HKU17 (icPDCoV-SHKU17) or the RBD of ISU73347 (icPDCoV-RBDISU) [79]. HKU17 and ISU73347, both sparrow deltacoronaviruses, had the closest phylogenetic relationship with PDCoV. Notably, both chimeric PDCoVs demonstrated decreased virulence and intestinal tropism loss in pigs; however, they retained the ability to infect the respiratory tract. Alhamo et al. found that the chimeric PDCoVs expressing S protein or the RBD of sparrow deltacoronavirus demonstrated lower replication ability in DF-1 cells and poultry than wildtype PDCoV; this result confirmed that PDCoVs, not sparrow deltacoronavirus, exhibit higher adaptability for a cross-species infection [80]. Taken together, these findings suggest that in CoV S protein determines their cell and tissue tropism and some key resides determine their cellular adaptability.

### 3.4. Screening Antiviral Drugs for Swine Coronaviruses

Considering the high pathogenicity of most emerging and reemerging swine CoVs, many scientists screened antiviral drugs that can inhibit pathogenic swine CoV replication. Edwards et al. synthetically constructed a recombinant SADS-CoV with a gene encoding tomato red fluorescent protein rather than ORF3 and applied it to understand whether remdesivir can efficiently inhibit SADS-CoV in vitro by calculating the florescence area [81]. In 2021, Li et al. constructed a recombinant PEDV that expresses nano luciferase (NLuc) to develop an NLuc-based, high-throughput screening platform to identify anti-PEDV compounds [82]. By using this platform, the authors screened 25 compounds—from a library of 803 natural compounds—that could significantly inhibit the replication of PEDV in vitro. Similarly, Fang et al. constructed a recombinant PDCoV that expresses NLuc by replacing the NS6 gene to identify antiviral drugs and found that PDCoV is sensitive to lycorine and resveratrol [83]. Chen et al. used the recombinant PEDV DR13 expressing the green fluorescent protein (GFP) to screen potential anti-PEDV drugs from carbazole alkaloid derivatives; the authors noted that three carbazole alkaloid derivatives exhibited high anti-PEDV activity [84]. Taken together, these studies indicate that reverse genetics systems are highly beneficial for antiviral research.

### 3.5. Determination of the Activity of Transcription Regulatory Sequences

Accessory and structural proteins are expressed by a series of nested subgenomic RNAs, which are regulated by transcription regulatory sequences (TRSs) [85]. The assembly of mature virions depends on the precise ratio of each viral structural protein. We Recently evaluated the transcriptional regulatory efficacy of PEDV TRSs by inserting an EGFP transcriptional unit between 3′UTR and the N gene of the PEDV genome using a PEDV reverse genetics system [86]. The results showed that, among all the inserted TRSs, the TRS of the M gene displayed the greatest ability to drive EGFP expression. The main research progress by each reverse genetics system is shown in Table 1.

## 4. Concluding Remarks

Despite the challenges associated with establishing the reverse genetics systems for swine CoVs, several research teams have generated the infectious cDNA clones of swine CoVs. The availability of reverse genetics systems has enabled precise genetic manipulation of the viral genome. The reverse genetics systems of swine CoVs have been used extensively; for instance, they have been used to investigate the biological function of CoV proteins, elucidate CoV transcriptional regulatory mechanism, screen antiviral drugs for CoVs, and identify virulence factors and cell tropism related to CoVs. The studies reported thus far deepen our understanding of the biology of swine CoVs and assist in the rational design of a new generation of vaccines with improved safety and efficacy for the emerging and re-emerging swine CoVs.

## Figures and Tables

**Figure 1 viruses-15-02003-f001:**
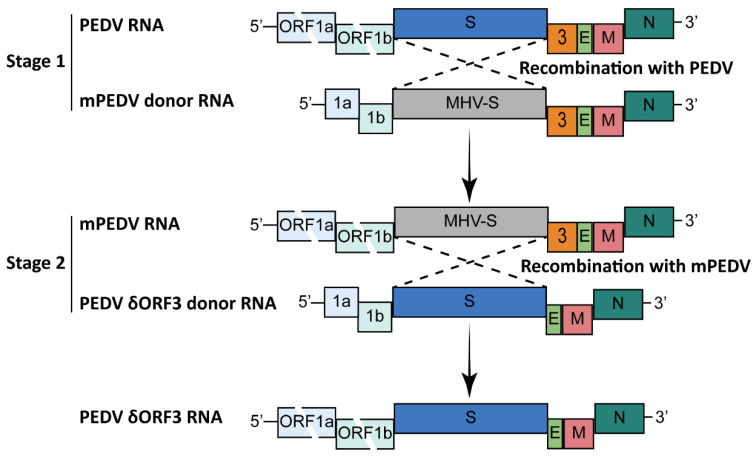
Targeted RNA recombination scheme to construct recombinant PEDV with ORF3 gene deletion [23]. MHV S transcripts with homologous arms of PEDV S gene are electroporated into PEDV-infected Vero cells. Then, the recombinant virus is obtained by plaque purification in murine L cells. Transcripts with expected mutations are electroporated into the recombinant virus-infected L cells; after 4 h post-infection, the recombinant virus with expected mutations can be obtained by plating the infected L cells onto monolayers of Vero cells.

**Figure 2 viruses-15-02003-f002:**
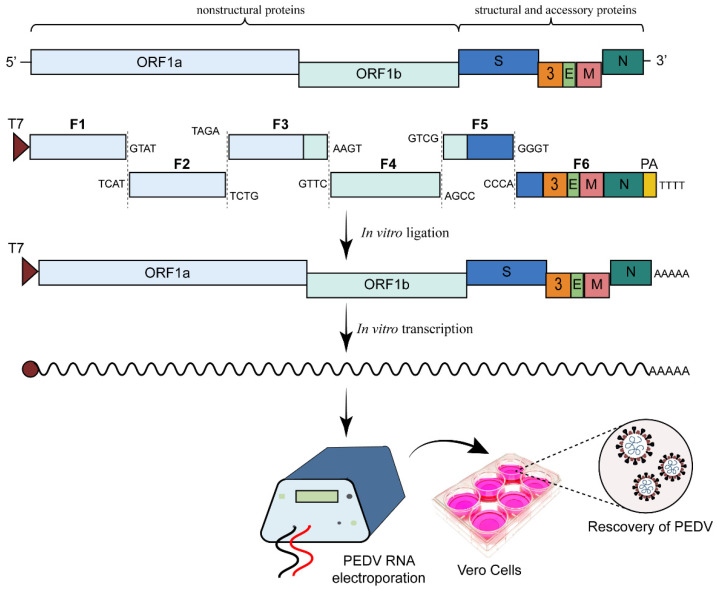
Engineering the genome of PEDV by in vitro transcription [33]. The genome structures of PEDV and the in vitro ligation approach are shown. Cohesive overhangs are shown; the length is not to scale. The full-length cDNA of PEDV is directionally assembled in vitro and then transcribed into infectious genomic mRNA with a T7 transcription kit. The genomic mRNA is electroporated into Vero cells to rescue the recombinant PEDV.

**Figure 3 viruses-15-02003-f003:**
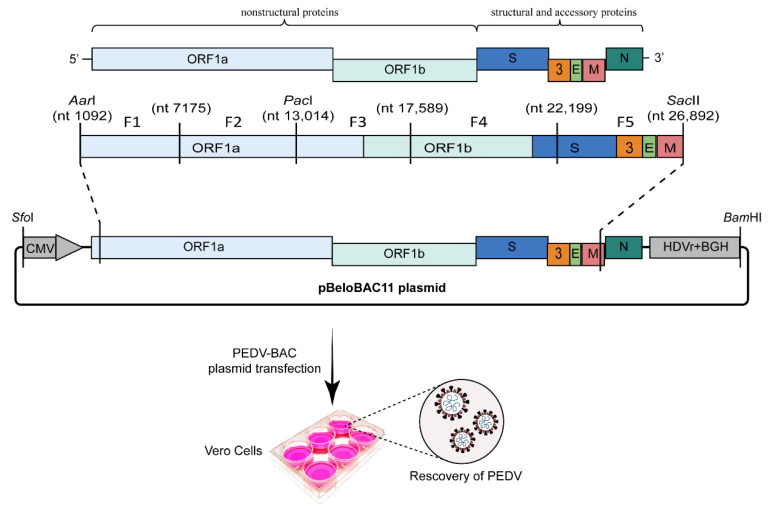
Flowchart for construction of the PEDV infectious clone using BAC system [39]. The genome structure of PEDV followed by DNA fragments amplified by RT-PCR is shown. The F1 and F2 DNA fragments are ligated to an intermediate plasmid pBAC-M-PEDV which cloned the CMV promoter, PEDV genome segments [nucleotide (nt) 1–1,092, *Pac*I cleavage site, followed by nt 22,199 to 3′ UTR] a poly(A) tail, HDVr, and BGH polyadenylation sequence by homologous recombination; similarly, F3, F4, and F5 are also ligated to pBAC-M-PEDV by homologous recombination. Finally, DNA fragments (F3–F5) were cut by restriction enzymes *Pac*I and *Sac*II and then ligated to the recombinant BAC plasmid cloned F1 and F2 with T4 DNA ligase to obtain an infectious PEDV clone plasmid. The virus could be rescued through transfection of the recombinant BAC plasmid into Vero cells.

**Figure 4 viruses-15-02003-f004:**
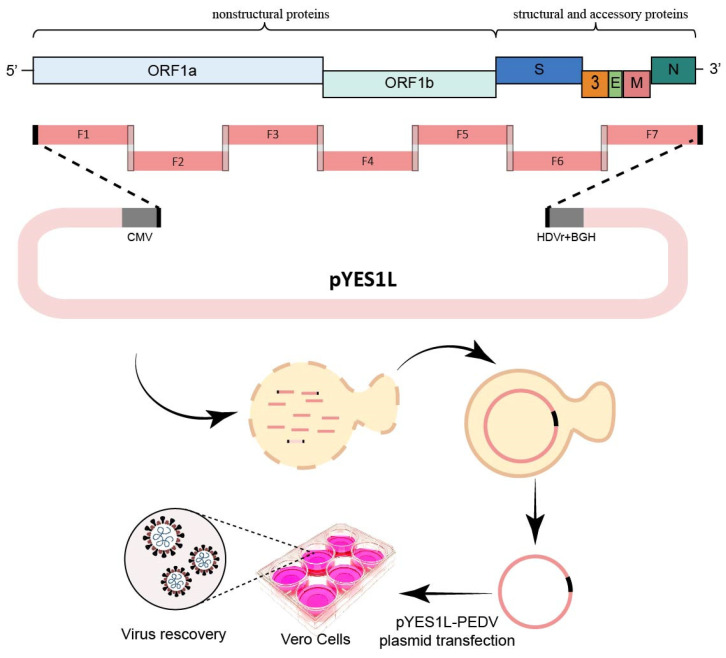
Engineering the genome of PEDV through yeast-based vector. The complete genome of PEDV was divided into seven fragments with at least 30 nt overlaps with neighbor fragments. All the seven cDNA fragments were transformed together with a linearized vector (pYES1L) into yeast competent cells for assembly through transformation-associated recombination in yeast. After identification and extraction of the positive clones, the full-length cDNA clones were transfected into Vero cells for virus recovery.

**Figure 5 viruses-15-02003-f005:**
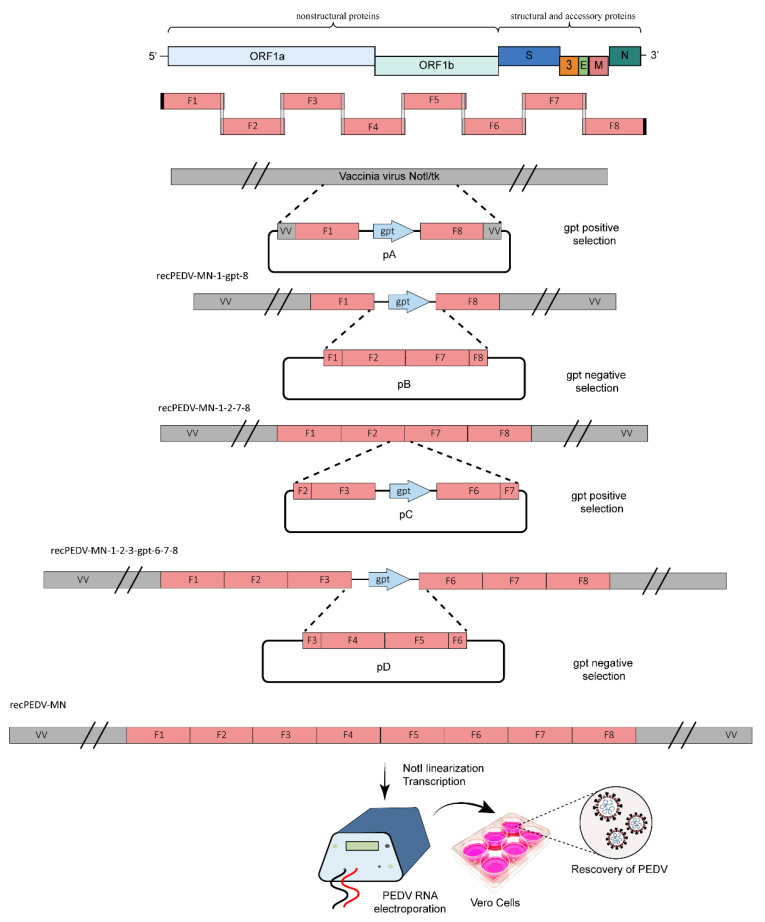
Vaccinia virus vector-based reverse genetics system for PEDV [43]. The PEDV genome was divided into eight fragments (F1–F8), which were then used to construct four plasmids: pA, containing fragment 1 upstream and fragment 8 downstream of the GPT gene; pB, containing fragment 2 and 7; pC, containing fragment 3 upstream and fragment 6 downstream of the GPT gene; pD, containing fragment 4 and 5. The genome was introduced into vaccinia virus genome through four rounds of vaccinia virus-mediated homologous recombination with GPT as a positive or negative selection marker. After linearization of the vaccinia virus genome by *Not*I digestion, the infectious mRNA was transcribed and electroporated into Vero cells to rescue recombinant PEDV.

**Figure 6 viruses-15-02003-f006:**
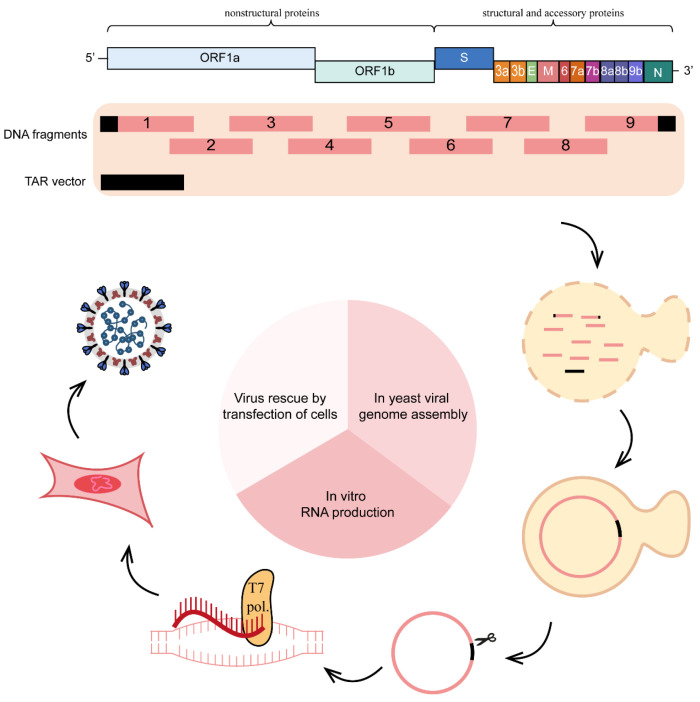
General workflow of TAR cloning for SARS-CoV-2 [44]. The genome of SARS-CoV-2 was divided into nine fragments with 45–500 base pair (bp) overlaps which were amplified by RT-PCR or overlapping PCR. The DNA fragments were co-transformed with TAR cloning vector into yeast for TAR cloning. The yeast plasmids were extracted from yeast culture and linearized. After being transcribed through in vitro transcription, the genomic mRNA was electroporated into susceptible cells for recovery.

**Figure 7 viruses-15-02003-f007:**
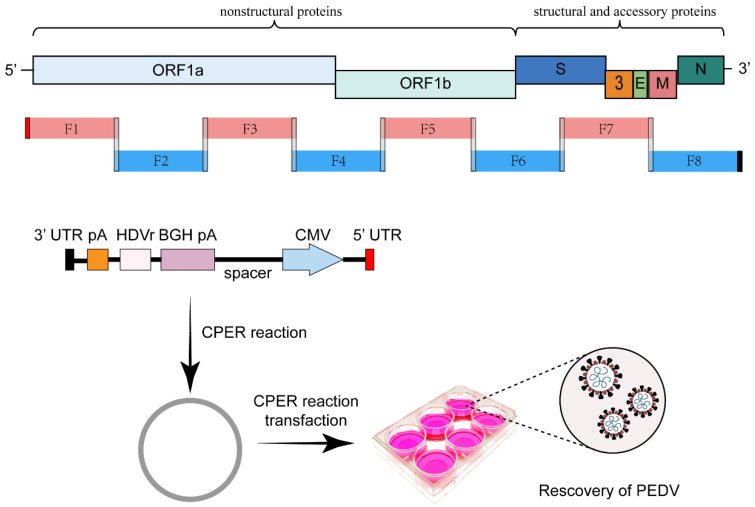
Scheme of CPER method to engineer the coronavirus genome. The complete CoV genome was covered by eight overlapping DNA fragments. A linker DNA fragment contains homologous arms, CoV 3′ UTR, a poly(A) tail, HDVr, a BGH polyadenylation sequence, a spacer sequence, a CMV promoter sequence, and CoV 5′ UTR. The linker sequence and overlapping DNA fragments were assembled through CPER, and, then, the CPER mixture was transfected into susceptible cells for virus recovery.

**Table 1 viruses-15-02003-t001:** Main research progresses using each reverse genetics system.

Types of Reverse Genetics System	Main Research Progresses	References
Targeted RNA recombination	The ORF3 is not essential for PEDV replication; the role of cellular adaptation of PEDV S2 in Vero cells	[23]
In vitro ligation method	The 197-amino-acid of S1 is a virulent factor in PEDV; YxxF motif of S2 subunit is associated with PEDV virulence; nonstructural proteins (nsp1, nsp15, and nsp16) are virulence factors for PEDV; accessory protein, NS6, is the virulence factor for PDCoV; PEDV nsp14 determines the viral genetic stability; Screening antiviral drugs for SADS-CoV	[48,50,51,53,54,55,81]
BAC-based ligation	Screening antiviral drugs for PDCoV; E protein is associated with PEDV pathogenicity; the seven-amino-acid motif is associated with the PEDV virulence; determined the regulatory activity of PEDV TRSs; PEDV can antagonize the immune suppression of C3 via inhibiting nsp1 phosphorylation of CEBP/β; trypsin-determinant via PEDV S2	[58,59,62,83,86]
Yeast-based method	The ORF3 is not essential for PEDV replication	[42]
Vaccinia virus-based recombination	The role of ORF3 and S in the pathogenicity of PEDV	[43]

## Data Availability

Not applicable.

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
