# Peer review of "Reverse Genetics Systems for Emerging and Re-Emerging Swine Coronaviruses and Applications"

_viruses, 2023, doi:10.3390/v15102003_

Round 1

Reviewer 1 Report

The authors summarize the established reverse genetic systems for the emerging and re-emerging swine coronaviruses and their advantages and disadvantages of these systems. This review also described the applications of these systems in elucidating virus-related pathogenesis, developing vaccine candidates, screening antiviral drugs, et al. Generally, this review is well written, with clear figures and proper references. However, there are several questions needing to be addressed before acceptance. But, in general, it is an excellent review, which will be interested for the broad scientific community.

Specific comments:

1. In line 25, please delete the “at the DNA level”, Because targeted RNA recombination method to manipulate the genome was performed at the RNA level, so this description was not so precise.

2. Most of the figures are missing the detail descriptions in the figure legend, please supplement the necessary information in figure legend.

3. In line 71, the sentence should be revised as “PDCoV was first identified from fecal samples of pigs in a surveillance study in Hong Kong in 2012”.

4. Some grammar errors are found in the manuscript, such as a comma should be added after “stability of its cloned genes” in line 150; the word “in” in line 203 should be revised as “of”.

5. In line 239, “can be” should be revised as “was”.

6. In line 242, “overlaps to” should be revised as “overlapping”.

7. In line 333, “titers of” should be added before IgG and IgA, and “stronger” should be added before protection effects.

8. In line 335, if the paper has been published online, the citation should be added in the manuscr

Minor editing of English language required

Reviewer 2 Report

This review summarizes the current knowledge regarding the major reverse genetics systems (targeted RNA recombination, in vitro ligation, BAC-based ligation, vaccinia virus vector-based, and yeast-based reverse genetics system) designed and used to study swine coronaviruses. Although such summary is of interest, the review fails for the most part to provide a conceptual and critical assessment/analysis of each system advantage and disadvantage, while much attention is given to insignificant/minor technical details. This is a major flaw, and the review needs to be carefully revised throughout to correct it. Also, the manuscript would benefit from including a table or tables summarizing main advancements made using each approach. The provided visuals are excellent and very useful.

The language/grammar/style need careful editing (ideally by a native English speaker) throughout. SOME (NOT ALL) examples of poor grammar, awkward phrasing and inaccuracies/typos are summarized below.

P2, LL22-23: “Currently, there are five reverse genetic approaches have been successfully applied to engineer the genome of the swine coronavirus, including target RNA..” → “Currently, there are five reverse genetic approaches that have been successfully applied to engineer the genome of the swine coronavirus, including target RNA..”

P2, LL23-24: “..target RNA recombination, in vitro ligation method, BAC-based ligation method, vaccinia virus vector-based reverse genetic system, and yeast-based method” → “..targeted RNA recombination, in vitro ligation method, BAC-based ligation, vaccinia virus vector-based reverse genetic system, and yeast-based reverse genetics system”. Please revise throughout to say ‘reverse genetics system’ not ‘reverse genetic system’

P3, LL34-35: “Coronaviruses (CoVs) are the largest positive single-strand RNA viruses under the family of Coronaviridae with four genera, Alphacoronavirus, Betacoronavirus, Gammacoronavirus, and Deltacoronavirus.” → “Coronaviruses (CoVs) are the largest positive single-stranded RNA viruses that belong (or within) the family of Coronaviridae consisting of four genera, Alphacoronavirus, Betacoronavirus, Gammacoronavirus, and Deltacoronavirus.”

P3, LL38-39: “Two overlapping ORFs, ORF1a and ORF1b, encode the polymerase proteins, which are further cleaved into nonstructural proteins (nsps).” → “Two overlapping ORFs, ORF1a and ORF1b, encode the polymerase complex proteins, which are further cleaved into nonstructural proteins (nsps).”

P3, LL49-50: “PEDV, SADS-CoV, and PDCoV are considered emerging or re-emerging swine coronavirus, since TGEV, PRCV, and PHEV have infected the swine for decades” → P3, LL49-50: “PEDV, SADS-CoV, and PDCoV are considered emerging or re-emerging swine coronaviruses, while TGEV, PRCV, and PHEV have been circulating in swine populations for decades”

P4, LL58-59: “Later, a high prevalence of PDCoV infection in swine was reported in the United States and China, which caused huge economic losses to the pig industry [17, 18].” PDCoV has not been associated with huge economic losses. Please rephrase to tone down, or cite a appropriate reference to support this statement. The ones that are currently cited do not report on PDCoV-associated HUGE economic losses.

P4, L65: “Viral reverse genetics is the purposeful modification of the genome at the viral genome level” → “Reverse genetics of RNA viruses refers to the purposeful modification of the viral genome”     

P8, LL129-130: “The remaining genome of PEDV AJ1102 was divided 129 into five fragments and amplified by RT-PCR ” → “The remaining genome of PEDV AJ1102 was amplified in five fragments using RT-PCR ”

P15, LL213-214: “The circular polymerase extension reaction (CPER) based method has recently been an attractive method to rescue recombinant positive RNA viruses.” → “The circular polymerase extension reaction (CPER) based method has recently become an attractive technique to rescue recombinant positive RNA viruses.” Also, please provide a reference to support this statement.

P17, LL246-247: “In 2018, Wang et al. applied the reverse genetic systems of two PEDV strains to confirm that the S gene is not the only virulent factor for PEDV [47].” → “In 2018, Wang et al. applied the reverse genetic systems of two PEDV strains to confirm that the S gene is not the only virulence factor for PEDV [47].”

P17, L248: “Except for the structure proteins, nonstructural proteins, including nsp1, nsp15, and nsp16, are also linked..” First, it should be ‘structural proteins’, and not ‘structure proteins’. Second, it should be ‘Besides’ not ‘Except for’ → “Besides the structural proteins, nonstructural proteins, including nsp1, nsp15, and nsp16, are also linked..”

P18, L272: “For example, our study recently proved that the carboxy-terminal of the S2 subunit contributes to…” → “For example, our recent study demonstrated that the carboxy-terminal of the S2 subunit contributes to…”

P19, L286-287: “Moreover, the reverse genetic system also has the potential to be the expression vectors that deliver antigens with interest.” → “Moreover, the reverse genetic system also has the potential to be the expression vectors to deliver antigens of interest.”

P20, LL300-301: “Currently, the isolation of wild PEDV in vitro…” → “Currently, isolation of wild-type (or field) PEDV strains in vitro…”

P21, LL316-317: “..which demonstrated 316 that the PDCoV not sparrow deltacoronavirus exhibits..” →  “..which demonstrated that the PDCoV but not sparrow deltacoronavirus exhibits..”

P21, L321: “The emerged or reemerged swine coronaviruses are pathogenic coronaviruses..” → “Emerging or reemerging swine coronaviruses are pathogenic coronaviruses..”

Pleasee see details in the report
